

# Determinants of behavioural and biological risk factors for cardiovascular diseases from state level STEPS survey (2017–19) in Madhya Pradesh

Arun M. Kokane[1], Rajnish Joshi[2], Ashwin Kotnis[3], Anirban Chatterjee[1], Kriti Yadav[1], G Revadi[1], Ankur Joshi[1], Abhijit P. Pakhare[1]

[1] Community and Family Medicine, All India Institute of Medical Sciences, Bhopal, Bhopal, Madhya Pradesh, India
[2] General Medicine, All India Institute of Medical Sciences, Bhopal, Bhopal, Madhya Pradesh, India
[3] Biochemistry, All India Institute of Medical Sciences, Bhopal, Bhopal, Madhya Pradesh, India

## ABSTRACT

**Background**. National and statewide assessment of cardiovascular risk factors needs to be conducted periodically in order to inform public health policy and prioritise allocation of funds, especially in LMICs. Although there have been studies from India which have explored the determinants of cardiovascular risk factors, they have mostly been from high epidemiological transition states. The present study assessed the determinants of cardiovascular (CVD) risk factors in a low epidemiological transition state (Madhya Pradesh) using the WHO STEPwise approach to surveillance (STEPS).

**Methods**. A total of 5,680 persons aged 18–69 years were selected from the state of Madhya Pradesh through multi-stage cluster random sampling. Key CVD risk factors we sought to evaluate were from behavioural (tobacco, alcohol, physical activity, diet) and biological domains (overweight or obese, Hypertension, Diabetes, and Raised serum cholesterol). Key socio-demographic factors of interest were the caste and tribe groups, and rural vs urban location, in addition to known influencers of CVD risk such as age, gender and education level

**Results**. Those belonging to the scheduled tribes were more at risk of consuming tobacco (OR 2.13 (95% CI [1.52–2.98]), and a diet with less than five servings of fruits and vegetables (OR 2.78 (95% CI [1.06–7.24]), but had had the least risk of physical inactivity (OR 0.31 (95% CI [0.02–0.54]). Residence in a rural area also reduced the odds of physical inactivity (OR 0.65 (95% CI [0.46–0.92])). Lack of formal education was a risk factor for both tobacco consumption and alcohol intake (OR 1.40 (95% CI [1.08–1.82]) for tobacco use; 1.68 (95% CI [1.14–2.49]) for alcohol intake). Those belonging to schedules tribes had much lower risk of being obese (OR 0.25; 95% CI [0.17–0.37]), but were at similar risk of all other clinical CVD risk factors as compared to other caste groups.

**Conclusion**. In the current study we explored socio-demographic determinants of behavioural and biological CVD risks, and found that in Madhya Pradesh, belonging to a scheduled tribe or living in a rural location, protects against being physically inactive or being overweight or obese. Increasing age confers a greater CVD-risk in all domains. Being a male, and lack of formal education confers a greater risk for behavioural domains, but not for most clinical risk domains. Future efforts at curbing CVDs should

be therefore two pronged –a population-based strategy targeting biological risk factors, and a more focussed approach directed at those displaying risky behaviour.

## INTRODUCTION

Cardiovascular diseases (CVDs) are responsible for more than a quarter of all deaths in India, and this proportion has been on the rise since last two decades (*Prabhakaran et al., 2018*). Contribution of CVDs to DALYs has also increased from 6.9% in 1990, to 14.1% in the year 2016 (*Prabhakaran et al., 2018*). This rise has enormous implications for the health system, as well as for the overall social and economic fabric of the country (*Mahal, Karan & Engelgau, 2010*; *Thakur et al., 2011*). Distribution of CVD risk factors is heterogenous across different regions and states of India (*Dandona et al., 2017*). States like Kerala, and Punjab have a higher CVD-risk factor prevalence and are termed as high Epidemiological Transition Level states (ETLs), in contrast to Madhya Pradesh (MP) which is a low epidemiological transition state. Madhya Pradesh has lower proportion of individuals who consume tobacco, and who are overweight or obese as compared to high epidemiological transition states. The prevalence of hypertension and diabetes is also low in MP as compared to other states (*Kokane et al., 2020*).

The lower prevalence of CVD risk factors in MP may be due to the fact that it is lower down in the epidemiological transition curve;in the absence of any corrective interventions, it can be assumed that it may start resembling the other high ETL states, in which case the distribution of CVD risk factors is likely to be uniform across various societal caste subgroups, or rural vs urban locations. The lower prevalence of CVD risk factors can be hypothesised to be due to a number of factors unique to the state—a significant proportion of the state population is rurally situated—a factor known to be predictive of lesser CVD risk (*Geldsetzer et al., 2018*). The state also has a substantial tribal population—almost one-fifth of the state population is tribal (*Census of India: Tables on Individual Scheduled Castes (SC) and Scheduled Tribes (ST), 2011*). A substantial proportion—almost three-fourth—of the tribal population in MP also reside in the rural areas, which has been shown to be associated with lesser prevalence of CVD risk factors (*Kapoor et al., 2014*).

These assertions are however, debatable. Studies from rural India have shown an increasing trend of CVD risk factors—almost similar to urban India (*Bhadoria et al., 2017*; *Tushi et al., 2018*). Multiple studies in tribal populations in India have also had contrary findings. Multiple studies conducted in tribal regions in various parts of the country have found that CVD risk factor prevalence in the studied tribal communities was comparable or even higher as compared to estimates for general population (*Kandpal, Sachdeva & Saraswathy, 2016*; *Kshatriya & Acharya, 2019*; *Sathiyanarayanan, Muthunarayanan & Devaparthasarathy, 2019*). However, most of these studies have been limited by their

restricted geography, and comparisons with general population in other external studies, conducted in different time-frames.

We conducted this study to identify risk factors of NCDs and explore their distribution across socio-demographic factors. Key CVD risk factors we sought to evaluate were from behavioural (tobacco, alcohol, physical activity, diet) and biological domains (overweight or obese, Hypertension, Diabetes, and Raised serum cholesterol). Key socio-demographic factors of interest were the caste and tribe groups, and rural vs urban location, in addition to known influencers of CVD risk such as age, gender and education level. We aim to test the above hypothesis of skewed vs uniform distribution of CVD risk factors for the State of Madhya Pradesh.

## METHODS

### Primary data collection

We conducted a cross-sectional community-based survey based on WHO STEPS wise approach in ten districts of Madhya Pradesh, a state with a population of about 72.6 million, 15.34 million of which belongs to the scheduled tribes. We used a multi-stage cluster random sampling method for the purpose of our study. Firstly we selected all 10 divisions from within the state, followed by one district selected randomly from within each division. In the third stage, 100 Primary Sampling Units (PSUs)/clusters were selected by Probability Proportionate to Size (PPS) method, of which 70 were from rural and 30 from urban areas. A total of 5,680 participants between age of 18–69 years were interviewed and behavioural risk factors were assessed (STEP-1). Then anthropometry and assessment of blood pressure was done in 4,985 individuals (STEP-2); and point-of-care testing for blood glucose and cholesterol were done in 4,698 individuals (STEP-3). The blood glucose and cholesterol in the study participants was measured using Aina Blood monitoring system (Jana Care Inc, Boston (MA)) at the study site. The Point of care testing system is based on immunoassay based real time measurement of multiple analytes including blood glucose and cholesterol with report provided on a smart phone based device with tablet based portable system. In general, the test results are based on the instrument reading light reflected off a test strip that has changed color after blood has been placed in it. The darker the colour on the strip, the higher the analyte concentration. The procedure for estimation of lipids and glucose is based on Glucose oxidase-peroxidase method and Trinder method respectively. The reference range of glucose was per standard guidelines. All the data was collected digitally using mobile phone-based applications, and securely stored prior to analysis. The methods of primary data collection, and overall results are described in detail elsewhere (*Kokane et al., 2020*). Briefly, field investigators calculated the sampling interval for household selection in a cluster by dividing the total number of households in that cluster by 58. This was followed by random selection of first household, and selection of eligible individual from the household using logic based on Kish grid method which was inbuilt in the data collection tool. Subsequent households were chosen by adding sampling interval to the current household (serialised as 1), till 58 eligible individuals were selected from the cluster.

## Data analysis

All data analysis was done in IBM SPSS- 26 software (IBM SPSS-26 for Macintosh, Armonk, NY, USA) and graphs were created by using *ggplot2* package in R software. This was a multistage cluster survey and thus sampling weights were derived considering probability of selection at each STEP. Therefore Complex Sampling Plan was generated assigning strata, primary sampling unit and appropriate sampling weight. We have used standard definitions of behavioural and biological risk factors. All these variables were nominal variables. Determinants considered for behavioural risk factors were age group, gender, educational status, marital status, caste group/social class and locality of residence (urban/rural). Weighted prevalence along with its 95% confidence interval was estimated by using Complex Sample Frequencies command for each of behavioural and biological risk factors across determinants enlisted above. Similarly for biological risk factors (overweight, raised blood pressure, raised cholesterol and raised blood glucose) we have included behavioural risk factors (tobacco use, alcohol use, physical inactivity and low intake of fruits and vegetables) as determinants. Weighted logistic regression procedure was used to identify independent determinants of behavioural and biological risk factors. Appropriate reference subgroup was assigned and odds ratios along with their 95% confidence interval were recorded.

## Ethics issues and permissions

Institutional Human Ethics Committee (IHEC) of AIIMS Bhopal, India, approved this study (IHEC-LOP/2014/EF0018 Dated 30th Jan 2015) and Director Health Services, Public Health Department, Government of Madhya Pradesh provided field permissions for implementation of the study.

## RESULTS

The primary data collection was performed in 2018-19. On an average the respondents were middle aged (mean age 40.4 years), had low levels of education (mean years of education 4.6 years), and more than a quarter of them belonged to scheduled tribes ($n = 1628$, 28.7%). An overwhelming majority (85.1%) of the participants were married at the time of the survey, and 94.9% of the participants were employed. Overall 34.2% (95% CI [31.5–36.9]) individuals consumed tobacco products and 9.4% (95% CI [7.9–11.0]) did so in its smoked form. The proportion of individuals who currently consume alcohol was 4.5% (95% CI [3.7–5.5]%); those with insufficient physical activity was 19.6% (95% CI [17.0–22.5]%); and those who consumed less than 5 servings of fruits and vegetables was 98.5% (95% CI [97.9–98.9]%). A total of 18.7% (95% CI [15.4–22.5]) were either overweight or obese, 22.3% (95% CI [20.5–24.1]) had raised blood pressure, 6.8% (95% CI [5.6–8.3]) had raised blood glucose, and 4.3% (95% CI [3.4–5.3]) had raised cholesterol levels. The detailed overall results are described in a previous publication (*Kokane et al., 2020*).

The distribution and risk of four behavioural CVD risk factors (tobacco consumption, alcohol consumption, physical inactivity, and consumption of less than 5 servings of fruits and vegetables) was different across key sociodemographic subgroups. Increasing age, and male gender significantly increased the risk of tobacco consumption, alcohol consumption,

and being physically inactive. As compared to 18-30-year olds, risk of tobacco and alcohol use was almost two to three times higher in those 60 years and above (OR 1.99 (95% CI [1.49–2.65]) for tobacco; 2.78 (95% CI [1.39–5.54]) for alcohol intake). As compared to women, men had eight times higher risk of tobacco use (OR 8.84 (95% CI [6.98–11.20]) and fifty times higher risk of alcohol use (OR 53.74 95% CI [28.43–101.57]). Lack of formal education, a proxy indicator of socioeconomic status, was a risk factor for both tobacco consumption and alcohol intake (OR 1.40 (95% CI [1.08–1.82] for tobacco use, and 1.68 (95% CI [1.14–2.49]) for alcohol intake). Those belonging to the scheduled tribes were more at risk of consuming tobacco (OR 2.13 (95% CI [1.52–2.98]), and a diet with less than 5 servings of fruits and vegetables (OR 2.78 (95%CI [1.06–7.24]), but had had the least risk of physical inactivity (OR 0.31 (95% CI [0.02–0.54])). While residence in an urban area increased risk of alcohol consumption, residence in a rural area increased the odds of a fruit-poor diet, but reduced the odds of physical inactivity (OR 0.65 95% CI [0.46–0.92]) (Fig. 1, Table 1 and Table S1).

The distribution of four biological CVD risk-factors (Elevation in body mass index (BMI), raised blood pressure (BP), raised blood sugars, and on therapy for dyslipidaemia or elevated cholesterol levels) was less heterogenous, as compared to the behavioural risks. Increasing age was a significant risk factor for all the biological CVD risk factors. As compared to 18–30 year olds, those 60 years and above were at a three times greater risk of being overweight (OR 3.07; 95% CI [2.11–4.46]) or elevated sugars (OR 2.96; 95% CI [1.72–5.10]), four times greater risk of elevated cholesterol levels (OR 4.39; 95% CI [2.61–7.36]), and more than ten times greater risk of having an elevated blood pressure (OR 13.93; 95% CI [9.78–18.32]). Compared to men, women had a higher risk of being overweight or obese (OR 1.88; 95% CI [1.49–2.37]), and having elevated cholesterol levels (OR 1.72; 95% CI [1.26–2.36]). Men were at a higher risk for having diabetes mellitus or elevated blood sugars (OR 1.71; 95% CI [1.27–2.30]). Higher education levels, and residence in an urban area increased the risk of being overweight (OR 1.6; 95% CI [1.28–2.0]). Those belonging to schedules tribes had much lower risk of being obese (OR 0.25; 95% CI [0.17–0.37]), but were at similar risk of all other biological CVD risk factors as compared to other caste groups (Fig. 2, Table 2 and Table S1).

## DISCUSSION

In the current study we explored socio-demographic determinants of behavioural and biological CVD risks, and found that in Madhya Pradesh, belonging to a scheduled tribe or living in a rural location, protects against being physically inactive or being overweight or obese. Increasing age confers a greater CVD-risk in all domains. Being a male, and lack of formal education confers a greater risk for behavioural domains, but not for most biological risk domains. Distribution of behavioural risk factors such as tobacco use, smoking, alcohol use, and physical inactivity is heterogenous across age, gender, educational status, caste, and locational subgroups. Distribution of biological risk factors such as hypertension, diabetes mellitus, elevated cholesterol however is heterogenous only for age and gender, but is quite homogenous for education status, caste, and locational subgroups.
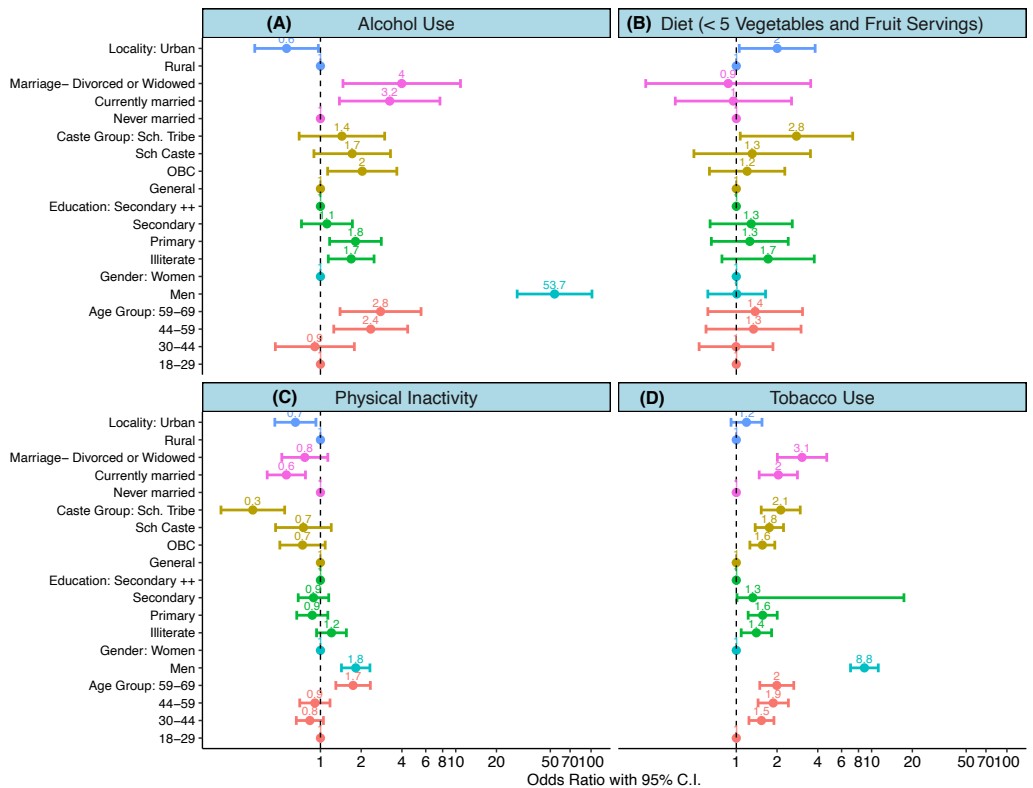

**Figure 1** (A–D) Determinants of behavioural risk factors. (A-D) show results of weighted logistic regression analysis wherein each of the behavioural risk factor was dependent variable. An odds ratio with its 95% confidence interval are shown for each of the determinants. Colours indicate category of each determinant. Scales are logarithmic.

There is an apparent dichotomy and paradox in the determinants of behavioural and biological risk factor. Behavioural risk factors are believed to be precursors of biological risk, that subsequently lead to more CVD events. Given this pathway, a rise in level of biological risk should be expected in the coming decades. Physical inactivity, being overweight, and having an elevated cholesterol level are expected to belong to different time-points on the same pathway. However we found that despite being more active, and having normal weight, those living in rural locations had serum cholesterol comparable to that in urban areas. Proportion of individuals with elevated sugars was also similar in urban and rural areas, and this was despite a marginally better intake of fruits and vegetables in the rural areas. Recent studies have placed a greater reliance on metabolic vulnerability for CVD outcomes, as compared to obesity alone (*De Rooij et al., 2016*). These findings suggest that health-systems need to aim for implementation of CVD-risk reduction strategies in all high-risk populations, including seemingly healthy non-obese but vulnerable individuals in rural areas, or belonging to scheduled tribes.

As compared to recent state-wide STEPS surveys in high epidemiological transition states such as Punjab, Haryana, and Kerala (*Thakur et al., 2016*; *Thakur et al., 2019*; *Sarma et al., 2019*), the population in Madhya Pradesh smokes less, has a lower body-mass index,

**Table 1** **Prevalence of behavioural risk factors stratified by sociodemographic determinants.** Prevalence of behavioural risk factors along with its 95% confidence interval stratified by various socio-demographic variables is shown.

| Variable | Tobacco users (current oral or smokers i.e., in last 30 days) | Smokers (current smokers i.e., in last 30 days) | Alcohol users (drank alcohol in past 30 days) | Physical inactivity (<150 min of moderate-intensity activity per week, or equivalent) | Unhealthy diet (ate less than 5 servings of fruit and/or vegetables on average per day) |
|---|---|---|---|---|---|
| **Age Group** | | | | | |
| 18–29 | 26.1 (22.8–29.6) | 4.1 (2.9–5.8) | 2.4 (1.4–3.9) | 20.6 (17.1–24.6) | 98.4 (97.5–98.9) |
| 30–44 | 36.8 (33.5–40.3) | 11.1 (9.1–13.6) | 6.1 (4.9–7.5) | 16.6 (14.0–19.6) | 98.3 (97.3–99.0) |
| 44–59 | 41.6 (38.2–45.0) | 14.6 (12.1–17.5) | 5.7 (4.3–7.5) | 18.9 (15.9–22.4) | 98.8 (97.9–99.4) |
| 59–69 | 42.1 (38.0–46.4) | 12.5 (10.1–15.3) | 4.3 (3.0–6.1) | 32.6 (28.5–37.1) | 98.9 (97.8–99.4) |
| **Gender** | | | | | |
| Men | 61.0 (57.5–64.4) | 24.3 (21.0–27.9) | 11.8 (9.7–14.2) | 25.3 (21.7–29.2) | 98.5 (97.7–99.0) |
| Women | 18.8 (15.8–22.1) | 0.8 (0.5–1.2) | 0.3 (0.2–0.6) | 16.3 (13.4–19.7) | 98.5 (97.9–98.9) |
| **Education** | | | | | |
| Illiterate | 34.4 (30.6–38.4) | 10.9 (9.1–13.0) | 4.7 (3.6–6.1) | 18.4 (14.9–22.5) | 99.1 (98.5–99.5) |
| Primary | 40.9 (36.8–45.1) | 12.4 (9.9–15.4) | 6.9 (5.0–9.4) | 17.1 (13.8–20.8) | 98.5 (97.4–99.1) |
| Secondary | 35.0 (31.2–39.0) | 9.0 (6.9–11.7) | 4.0 (2.7–5.7) | 18.3 (14.5–22.8) | 98.5 (97.3–99.1) |
| Higher Secondary ++ | 29.7 (26.2–33.4) | 6.2 (4.8–8.0) | 3.3 (2.4–4.6) | 23.1 (20.0–26.5) | 97.8 (96.6–98.6) |
| **Caste Group** | | | | | |
| General | 27.3 (24.1–30.8) | 11.0 (7.7–15.7) | 3.2 (1.9–5.4) | 28.9 (22.2–36.6) | 97.7 (96.1–98.7) |
| OBC | 31.7 (28.5–35.2) | 6.9 (5.4–8.8) | 2.6 (1.8–3.6) | 22.3 (18.9–26.0) | 98.1 (97.3–98.7) |
| SC | 34.8 (30.6–39.1) | 11.3 (8.9–14.2) | 6.4 (4.4–9.1) | 23.4 (18.6–29.0) | 98.3 (96.2–99.3) |
| ST | 39.9 (35.0–44.9) | 11.7 (9.3–14.7) | 7.0 (5.5–8.9) | 10.9 (7.5–15.5) | 99.4 (98.7–99.7) |
| **Marital Status** | | | | | |
| Never Married | 26.2 (21.3–31.6) | 2.8 (1.5–5.0) | 1.9 (0.9–4.1) | 31.9 (25.8–38.7) | 98.2 (95.6–99.2) |
| Currently Married | 35.0 (32.1–37.9) | 10.4 (8.8–12.2) | 5.0 (4.0–6.1) | 17.7 (15.1–20.5) | 98.5 (98.0–98.9) |
| Divorced or Widowed | 37.3 (31.8–43.1) | 6.4(4.3–9.5) | 2.8 (1.6–4.9) | 24.2 (19.3–29.7) | 98.8 (97.2–99.5) |
| **Locality** | | | | | |
| Urban | 36.9 (33.6–40.3) | 10.6 (8.8–12.8) | 4.5 (3.6–5.7) | 16.8 (13.8–20.3) | 99.0 (98.3–99.4) |
| Rural | 28.3 (24.6–32.4) | 6.6 (4.8–9.1) | 4.4 (2.9–6.9) | 25.6 (21.2–30.5) | 97.5 (96.3–98.3) |

and has a lower percentage of individuals with diabetes and hypertension. However, within the state there are population subgroups that have risk-factor levels comparable to high ETL states. Those living in the urban areas and those belonging to the upper caste have a similar proportion of smokers as in Punjab (*Thakur et al., 2016*), and similar proportion of individuals with hypertension as in Haryana or Kerala (*Thakur et al., 2019*; *Sarma et al., 2019*). Those belonging to the scheduled tribes, have only half the proportion of overweight or obese individuals, as compared to the entire state, and one-fourth as compared to those belonging to the upper castes. This population subgroup also has the least proportion of physically inactive individuals. However, this thinness-activity advantage does not translate into a favourable situation for either hypertension, diabetes mellitus, or dyslipidaemia as these proportions are similar across caste groups. One of the explanations for this could be

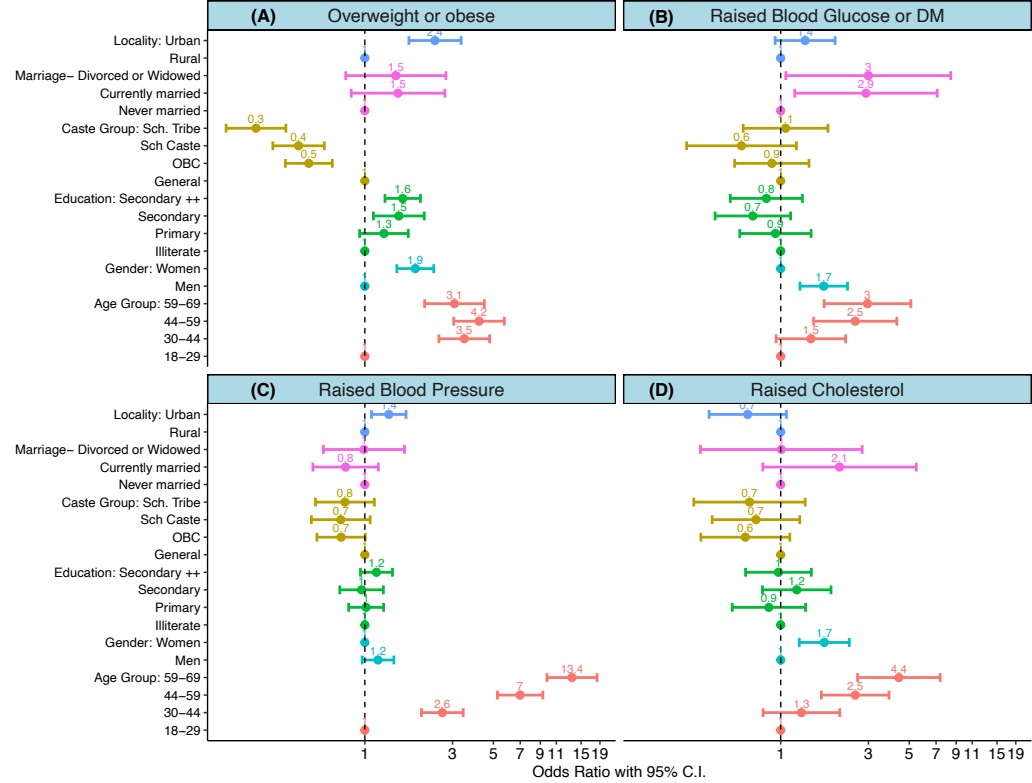

**Figure 2** (A–D) Determinants of biological risk factors. (A–D) show results of weighted logistic regression analysis wherein each of the biological risk factor was dependent variable. An odds ratio with its 95% confidence interval are shown for each of the determinants. Colours indicate category of each determinant. Scales are logarithmic.

high proportion of tobacco consumption or smoking amongst the scheduled tribes, which offsets favourable body-mass indices.

Current study provides an opportunity to ascertain if risk factors in scheduled tribes of Madhya Pradesh are different as compared to other tribal groups in the country. Previous studies in tribal populations of Himachal Pradesh (*Negi et al., 2016*), Uttarakhand (*Kandpal, Sachdeva & Saraswathy, 2016*), or Sikkim (*Sarkar et al., 2006*) have found a high proportion of individuals who were overweight or obese (38.2%, 56.6%, 13.8%, and 34.8% respectively). This contrasts with scheduled tribes in Madhya Pradesh where this proportion is only 7.3%. Proportion of the tribal population who had hypertension was found to be 20.1% in Madhya Pradesh, comparable to Kinnaur in Himachal Pradesh (19.7%) (*Negi et al., 2016*), and Toto tribe in West Bengal (18%) (*Sarkar et al., 2006*), who also lead similarly rural and agrarian lives. In contrast, proportion of tribal individuals with hypertension was 33.2% in Tripuris (*Sen et al., 2017*), 55.17% in Bhutias (*Sarkar et al., 2006*), and 43.4% in Rang Bhotias (*Kandpal, Sachdeva & Saraswathy, 2016*) all of whom are residing mostly in urban areas, probably because urban residence has an association with less physical activity, more obesity, and a relatively high levels of affluence (*Anjana et al., 2014*).

**Table 2  Prevalence of biological risk factors for NCDs stratified by sociodemographic variables and behavioural risk factors.** Prevalence of biological risk factors along with its 95% confidence interval stratified by various socio-demographic variables is shown.

| Variable | Overweight or Obese (BMI >= 25 kg/m2) $n = 4,985$ | Raised BP (SBP>140 or DBP>140 or taking antihypertensives) $n = 4,985$ | Raised Blood Glucose (Fasting blood glucose >126 mg/dL or taking drugs for diabetes) $n = 4,698$ | Raised cholesterol (Total cholesterol>190 mg/dl or on drug for cholesterol) $n = 4,698$ |
|---|---|---|---|---|
| **Age Group** | | | | |
| 18–29 | 6.4 (4.9–8.4) | 9.0 (7.5–10.8) | 3.5 (2.3–5.3) | 2.6 (1.7–3.8) |
| 30–44 | 19.9 (16.4–23.8) | 19.5 (17.1–22.1) | 6.4 (4.9–8.3) | 3.6 (2.7–4.8) |
| 44–59 | 21.7 (18.0–25.9) | 39.1 (35.6–42.7) | 11.0 (8.6–14.0) | 6.5 (4.8–8.6) |
| 59–69 | 17.2 (13.5–21.6) | 55.7 (51.0–60.2) | 13.2 (10.3–16.8) | 9.9 (7.3–13.2) |
| **Gender** | | | | |
| Men | 10.6 (8.7–12.9) | 23.5 (20.7–26.4) | 8.4 (6.5–10.8) | 3.1 (2.3–4.3) |
| Women | 17.9 (14.9–21.3) | 21.7 (19.7–23.8) | 6.0 (4.7–7.5) | 4.9 (3.9–6.2) |
| **Education** | | | | |
| Illiterate | 13.6 (11.2–16.3) | 27.1 (24.6–29.8) | 8.6 (6.5–11.4) | 5.5 (4.2–7.1) |
| Primary | 15.0 (11.8–19.0) | 21.3 (18.3–24.5) | 7.2 (5.2–9.8) | 3.5 (2.3–5.5) |
| Secondary | 15.0 (12.0–18.6) | 15.8 (13.3–18.5) | 4.7 (3.3–6.8) | 4.2 (2.9–6.0) |
| Higher Secondary + | 17.6 (14.1–21.7) | 21.6 (19.0–24.5) | 5.9 (4.5–7.8) | 3.4 (2.4–4.8) |
| **Caste Group** | | | | |
| General | 30.0 (24.4–36.2) | 30.3 (25.2–35.9) | 7.9 (5.2–11.7) | 6.2 (3.8–9.9) |
| OBC | 17.8 (14.6–21.4) | 22.3 (20.1–24.7) | 6.8 (5.4–8.6) | 4.0 (2.9–5.4) |
| SC | 14.9 (11.6–19.0) | 21.3 (17.9–25.2) | 4.8 (2.6–8.4) | 4.4 (3.1–6.2) |
| ST | 7.3 (5.6–9.5) | 20.1 (17.3–23.2) | 7.6 (5.2–11.0) | 4.1 (2.7–6.2) |
| **Marital Status** | | | | |
| Never Married | 5.4 (3.3–8.8) | 13.0 (9.6–17.3) | 1.8 (0.8–4.3) | 1.3 (0.5–3.1) |
| Currently Married | 16.3 (13.7–19.3) | 22.0 (20.2–23.9) | 7.3 (5.9–8.9) | 4.7 (3.7–5.8) |
| Divorced or Widowed | 19.1 (14.6–24.7) | 42.3 (36.3–48.5) | 9.8 (7.0–13.7) | 4.1 (2.5–6.7) |
| **Locality** | | | | |
| Rural | 10.7 (9.0–12.6) | 20.8 (18.8–22.9) | 6.5 (5.0–8.5) | 4.7 (3.6–6.1) |
| Urban | 25.6 (20.3–31.8) | 25.7 (22.5–29.1) | 7.6 (5.7–10.0) | 3.3 (2.3–4.7) |

Level of education attained by an individual is a key indicator of socio-economic status, and lower levels are generally associated with a greater risk (*Kaplan & Keil, 1993*). These risks have also traditionally been found to translate into more cardiovascular events, and a greater mortality (*Mensah et al., 2005*). In contrast, previous studies from India have reported that those with a higher education level or asset ownership have higher proportion of hypertension, diabetes mellitus, and obesity (*Ali et al., 2016*; *Corsi & Subramanian, 2019*). This finding has been described both in large cross-sectional studies (*Ali et al., 2016*; *Geldsetzer et al., 2018*) as well as in a recently reported cohort from various low- and middle-income countries (*Rosengren et al., 2019*). Despite the individuals with lower education having a better CVD risk profile, they have a higher mortality, attributed to worse access to health-care (*Rosengren et al., 2019*). Economically weaker nations have

greater health inequalities and an overall poor control of health-system dependent CVD risks, such as hypertension or diabetes mellitus (*Palafox et al., 2016*).

A key strength of the current study is its state-wide geo-cultural coverage. We attempted to decipher the complex framework of interactions and covariance in NCD occurrence with methodological robustness. The reference categories chosen in explanatory variables had a similar directional relationship with outcome variables. Moreover weighted techniques adopted in this study address possible sample heterogeneity (attributed to vast geographical and socio-cultural areas) and reveals the true covariate effect on outcome. However this frequentist approach may sometimes undermine the dispersion of data and thus may falsely augment the probability of statistical significance. Further, respondents could have experienced recall or information biases while responding to the questionnaire. The scope of the current study is to explore epidemiological relationships in a cross-sectional manner. This design limitation does not allow us to discern temporality amongst such complex risk-factors.

## CONCLUSION AND RECOMMENDATIONS

In the current study, we found behavioural factors to be heterogenous, suggesting that overall lower prevalence of CVD-risk in the state is likely to be due to less vulnerable population subgroups. In contrast biological risk factors were quite homogenously distributed, with lower prevalence as compared to other states in India, suggesting that 'so-called' less vulnerable groups in terms of behavioural risks are also at a significant biological risk. These findings call for different strategies for CVD-risk reduction in different population subgroups in the State.

The Health & Wellness Centres (HWCs) proposed under the Pradhan Mantri Jan Arogya Yojana were envisaged expand primary healthcare services for non-communicable diseases as well (*Lahariya, 2020*). These services includes screening of adults for non-communicable diseases through community health workers, then confirmation of NCD risk factors or disease at HWC and initiation of lifestyle modification measures and drug therapy etc These centres will cater to clusters of defined villages in their vicinity. Optimal utilization of HWCs could go a long way in providing a plethora of services targeting these NCD modifiable risk factors and improving the health and well-being of the people. These centres need to consider heterogeneity in distribution of risk factors in population sub-groups and prioritize their activities accordingly. Also, public health managers monitoring HWCs need to ensure equity in terms of logistics, supplies and other support considering disparities in distribution of NCD risk factors among HWCs.

### Funding

Indian Council of Medical Research, New Delhi (ICMR) funded this study under extramural grants. (IRIS No 2014-2668). The funders had no role in study design, data collection and analysis, decision to publish, or preparation of the manuscript.

## Grant Disclosures

The following grant information was disclosed by the authors:
Indian Council of Medical Research, New Delhi (ICMR): IRIS No 2014-2668.

## Competing Interests

Abhijit Pakhare is an Academic Editor of PeerJ.

## Author Contributions

- Arun M. Kokane and Rajnish Joshi conceived and designed the experiments, performed the experiments, analyzed the data, authored or reviewed drafts of the paper, and approved the final draft.
- Ashwin Kotnis conceived and designed the experiments, performed the experiments, authored or reviewed drafts of the paper, and approved the final draft.
- Anirban Chatterjee analyzed the data, authored or reviewed drafts of the paper, and approved the final draft.
- Kriti Yadav performed the experiments, prepared figures and/or tables, and approved the final draft.
- G Revadi analyzed the data, prepared figures and/or tables, authored or reviewed drafts of the paper, and approved the final draft.
- Ankur Joshi performed the experiments, analyzed the data, authored or reviewed drafts of the paper, and approved the final draft.
- Abhijit P. Pakhare conceived and designed the experiments, performed the experiments, analyzed the data, prepared figures and/or tables, authored or reviewed drafts of the paper, and approved the final draft.

## Human Ethics

The following information was supplied relating to ethical approvals (i.e., approving body and any reference numbers):

Institutional Human Ethics Committee (IHEC) of AIIMS Bhopal, India, approved this study (IHEC-LOP/2014/EF0018 Dated 30th Jan 2015).

## Field Study Permissions

The following information was supplied relating to field study approvals (i.e., approving body and any reference numbers):

Permission and facilitation for data collection at field sites were provided by Directorate Health Services, Satpura Bhavan, Madhya Pradesh.

## Data Availability

Raw data are available in Supplementary File.

## Supplemental Information

Supplemental information for this article can be found online at http://dx.doi.org/10.7717/peerj.10476#supplemental-information.

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
