# Peer review of "Determinants of behavioural and biological risk factors for cardiovascular diseases from state level STEPS survey (2017–19) in Madhya Pradesh"

_PeerJ, doi:10.7717/peerj.10476_

## Round 0.1 · original submission · Minor Revisions

As you will see, all reviewers found your work interesting and worthy of publication, but have suggested a few minor edits/revisions. Please address all point raised and carefully describe your responses and any manuscript alterations n a rebuttal letter.

Could you consider using multivariate analyses to combine domains to show which are more prominent factors in the study population? That may make an interesting adjunct.

I look forward to reading the revised paper.

·

Basic reporting

1. This is a generally well-written manuscript on an important public health issue of the current world including the Indian population.

2. The literature review and context provided are adequate

3. Data presented in a clear manner but facts need to be corrected 9see comments for Results section).

Experimental design

The methods described are inadequate. The following points need to be addressed:

1. Although the sampling procedure to recruit subjects has been cited (reference 5), there is nothing in the Methods section of this manuscript. It would be good to provide a brief description to make this article independent. Readers will be able to understand the procedure without reading the reference article;

2. the definition of diabetes in their original descriptive article (reference number 5) and the current one is different in respect to the cut-off point of blood glucose (126 or 200 mg); The two manuscripts provide two different measurement approaches, fasting and random. This will have a big implication on the results of this paper.

3. Table 2 defines hypertension as SBP>140 or DBP>140, which should be DBP>90.

Validity of the findings

The results are presented clearly. However, there are points of concern:

1. The characteristics of the study participants have been rightly given at the beginning of this section. However, the results are erroneous. All the numbers given here for age, education, and tribes are for the females given reference number 5. These should have been for both sexes. Summary information for marital status and employment could be given as has been given for age and education.

2. Subsequent description of findings is in fact directly reproduced from Table 4 of reference number 5. Hardly there is any description of risk factors presented on the concerned table here (Table 1). Please see the lines 86 to 93.

3. Table 3 is redundant of Figures 1 and 2. Table 3 could be easily avoided.

Additional comments

Considering the journal's readership and similarity of population characteristics, it would be good to have a discussion of similar articles from neighboring countries, Bangladesh, Nepal, Myanmar, Sri Lanka, and the Maldives.

·

Basic reporting

The STEPS survey is designed for NCD surveillance for comparison globally and within the country. This study used data from STEP wise approach survey conducted. It is suggested to change the title as “Determinants of behavioural and biological risk factors for cardiovascular diseases in Madhya Pradesh, India. A WHO STEP wise approach”.

Line 20: termed as high epidemiological transition states (ETLs)- This should be Epidemiological Transition Level states
“Line 42 We conducted this study to understand socio-demographic determinants of CVD risk.” The objective of STEPS may be to identify the risk factors on NCD and not to understand the determinants however, this scientific paper aimed to analyse the determinants. Therefore, need to rewrite the sentence.
It is suggested to use smokeless tobacco in place of oral tobacco users as this will confuse the readers. Similarly, change DRANK ALCHOL to excessive use of alcohol or alcohol use whichever is appropriate.
Language editing is critical in accepting the paper.

Experimental design

It is appropriate to include the detailed methodology such as sampling design, selection of the respondents etc to make it reader friendly. It is also suggested to provide operational definition of all terms used in this paper.
“Line 58: individuals (STEP-2); and laboratory testing for blood glucose and cholesterol were done in 4698”. Globally, STEPS following point of care instruments for all biometrics however, in your study it was mentioned that laboratory testing. This needs to be explained, that includes collection, storage and transportation of blood samples etc.. In addition, what are the methods used for measuring both blood glucose and lipid profile. Also suggested to include rationale for choosing this method. It is also suggested to include the rational for choosing this method for biometric in the methodology session.

Validity of the findings

“Line 167 Current study provides a unique opportunity to ascertain if risk factors in scheduled tribes of
Line “168 Madhya Pradesh are different as compared to other tribal groups in the country.”

The above statement by the author needs to be re-written as this is not an unique opportunity to ascertain the risk factors in scheduled tribes. This is applicable in all population.

Age classification needs to be revised to make it a standard by making same intervals and inclusive.

Suggested to use WHO nomenclature for risk factors as the study followed WHO STEPS protocol.

Under the education category one of the categories “not formally educated” this needs to be clarified. Are they literate or illiterate? A state like MP which is low in education indicators there needs to be a category named illiterate.

All tables should be incorporated with sample size (n).

When we consider the biological risk factors only age and gender are major determinants. Other social factors are less likely to influence the biological risk factors. The author needs to re-consider the analysis.

It is suggested use similar terms- like biological risk factors and behavioural risk factors across the paper. Some places it was mentioned that clinical risk factors.

The author tried to capture the determinants of risk factors for CVD however, for the programmatic/policy point of view there needs to be clearly indicating the need for improvement in certain factors. Therefore, it is suggested to segregate the analysis by modifiable and non-modifiable risk factors that will provide proper guidance to policy makers. To be more precise, if the paper focused more on modifiable risk factors that would through light in providing appropriate health promotion interventions in the state to improve the health of the people.
Conclusions and recommendations should be based on the findings and provide appropriate suggestions to policy improvements. It is proposed to review the current government interventions towards NCDs in the country, especially in MP. The government initiative focused on NCDs namely “Health and Wellness Centres” providing multiple services to address the modifiable risk factors. The author needs to review this and provide suggestion to improve the activities of centres.

Additional comments

A good attempt in finding out the factors associated with NCD risk factors among tribes in India, a has relevance in the present context. Appreciate the authors in selecting the policy relevance topic for the research.

Reviewer 3 ·

Basic reporting

Overall it is an interesting dataset and analysis. Write up is clear but descriptive. Perhaps authors could use multivariate analyses to combine domains to show which are more prominent factors in the study population.

Experimental design

Overall experimental design is good but Hypotheses could be written more clearly so that readers are able to see what is being tested.

Validity of the findings

Generally good details of the analyses carried out and results are presented in figures and table. Authors should make clear in their write up which categories are reference categories for odds ratios.

Additional comments

Overall a good write up, please make hypotheses more explicit and discussion should more focused to the hypotheses.

---

## Round 0.2 · accepted · Accept

Thank you for addressing the issues raised.